# One-Pot Access to Functionalised Malamides via Organocatalytic Enantioselective Formation of Spirocyclic β-Lactone-Oxindoles and Double Ring-Opening [note 1]

**DOI:** 10.3390/molecules29153635

**Published:** 2024-07-31

**Authors:** Alastair J. Nimmo, Kevin Kasten, George White, Julia Roeterdink, Aidan P. McKay, David B. Cordes, Andrew David Smith

**Affiliations:** EaStCHEM, School of Chemistry, University of St Andrews, St Andrews, Fife KY16 9ST, UK; an52@st-andrews.ac.uk (A.J.N.); kk47@st-andrews.ac.uk (K.K.); george.white@chem.ox.ac.uk (G.W.); jr275@st-andrews.ac.uk (J.R.); apm31@st-andrews.ac.uk (A.P.M.);

**Keywords:** malamides, organocatalysis, isothiourea, β-lactone ring-opening, oxindole ring-opening, asymmetric synthesis, [2+2] cycloaddition

## Abstract

Malamides (diamide derivatives of malic acid) are prevalent in nature and of significant biological interest, yet only limited synthetic methods to access functionalised enantiopure derivatives have been established to date. Herein, an effective synthetic method to generate this molecular class is developed through in situ formation of spirocyclic β-lactone-oxindoles (employing a known enantioselective isothiourea-catalysed formal [2+2] cycloaddition of C(1)-ammonium enolates and isatin derivatives) followed by a subsequent dual ring-opening protocol (of the β-lactone and oxindole) with amine nucleophiles. The application of this protocol is demonstrated across twelve examples to give densely functionalised malamide derivatives with high enantio- and diastereo-selectivity (up to >95:5 dr and >99:1 er).

## 1. Introduction

Malic acid (2-hydroxybutanedioic acid) is a readily available platform chemical that is widely used in the food and cosmetic industries [1,2,3,4]. It is readily available in an enantiomerically pure form and has been utilised as a versatile C(4)-synthetic building block for a range of natural product syntheses [5]. Amide derivatives of malic acid (generally known as malamides) are prevalent in nature [6,7,8,9,10] and are of significant biological interest, as they are isosteric to asparagine and aspartic acid and have been used against HCV infections [11] and as a constituent of agents that provide DNA binding to p53 [12]. They can also be used as metalating reagents [13,14] and serve as metalloprotease inhibitors [15,16,17,18,19,20] or as peptidomimetics (Figure 1A) [13,17,21,22,23]. As a representative example, DuPont Merck developed a chiral auxiliary-based route to a range of matrix metalloproteinase (MMP) inhibitors based upon the malamide motif, with the absolute configuration at C(2) essential for bioactivity [17]. Despite this interest, catalytic enantioselective protocols to access bespoke highly functionalised malamide derivatives with high stereocontrol are generally lacking but could be of widespread interest to the synthetic community.

Our proposed strategy to access a series of malamides was built upon the established precedent for β-lactones and oxindoles to undergo nucleophilic ring-opening under defined reaction conditions. β-Lactones are prevalent in medicinally active compounds and natural products [24,25,26,27,28,29] but are susceptible to ring-opening [30,31,32,33,34,35,36,37,38,39,40,41,42,43,44,45,46,47,48,49,50,51,52,53,54], decarboxylation [55,56,57], β-elimination [37,58], or acid-catalysed ring expansions [48,59,60,61,62,63,64,65,66]. Ring-opening of β-lactones to generate isolable compounds is a common strategy, typically employing alcohols, alkoxides or primary/secondary amines as nucleophiles. Ring-opening can occur either through the scission of the O-alkyl bond (via an S_N_2 pathway) or the scission of the O-acyl bond (via an addition elimination process). The preference for the major reaction pathway is dependent upon the nucleophile and the reaction conditions, with O-alkyl cleavage typically being favoured with hard nucleophiles under neutral or acidic conditions and O-acyl cleavage favoured under basic conditions with softer nucleophiles [37,66]. However, the inherent preference for O-alkyl or O-acyl bond scission can be altered by the strategic placement of substituents that introduce a steric bias to favour one reaction mode over the other (Figure 1B, left). In a similar fashion, isatins and oxindoles are common motifs within bioactive compounds [67,68,69,70,71,72,73,74,75,76,77,78,79,80,81] and are known to be susceptible to ring-opening. For example, the incorporation of electron-deficient N-substituents within isatins and oxindoles is known to promote facile ring-opening, with the literature conditions indicating they can be readily opened by alcohols, alkoxides, hydroxides and amines [82,83,84,85,86,87,88,89,90,91,92,93,94,95,96,97] to give the corresponding esters and amides (Figure 1B, right). Using this approach, substituents that contain pendant or masked amines and alcohol functionality have also been employed to generate pyrrolidones [98,99,100,101,102,103,104,105,106,107] and butyrolactones [108,109,110], respectively, via intramolecular ring-opening.

Building upon these precedents, this work develops a strategy to access stereodefined and densely functionalised malamides in enantiopure form. Over the last decade, the use of isothioureas as enantiopure Lewis base catalysts have expanded widely. In recent work, the isothiourea HyperBTM promoted the in situ formation of β-lactone spirocyclic oxindoles via a [2+2] cycloaddition of C(1)-ammonium enolates to N-alkyl isatin derivatives, with selective β-lactone ring-opening facilitating product isolation [30]. In this work, the use of N-acyl isatins facilitates an enantioselective route to spirocyclic isatin-derived β-lactones, with double ring-opening (of the β-lactone and oxindole motifs) revealing the linear and densely functionalised malamide structure, including two contiguous stereogenic centres (Figure 1C). To the best of our knowledge, while reports of selectively opening either β-lactones or (spiro)oxindoles [111,112,113,114] have been disclosed previously, this double ring-opening strategy has hitherto been unknown.

## 2. Results

Building upon the literature precedent, initial studies probed the hypothesis that changing from an N-alkyl-protected isatin to incorporating a conjugating electron-withdrawing N-substituent would facilitate double ring-opening after enantioselective [2+2] cycloaddition. Initially, an *N*-phenyl substituent was considered, with *N*-phenyl isatin treated under the previously optimised reaction conditions for the [2+2] addition of C(1)-ammonium enolates to N-alkyl isatins [11a], followed by the addition of benzylamine (Table 1, entry 1). Unfortunately, no malamide is observed, and only the ring-opening of the isatin starting material is observed (≈30% yield by ^1^H NMR spectroscopic analysis). Unfortunately, the installation of an N-C(O)NHPh substituent led to the resulting isatin starting material being insoluble under the reaction conditions, returning only the starting material (entry 2). While the use of *N*-Ts protected isatin led to a complex product mixture (entry 3), *N*-Boc-substituted isatin gave the desired malamide product 1 resulting from double ring-opening in a good 67% yield and with excellent stereocontrol (entry 4, >95:5 dr, 99:1 er) when 3.0 equivalents of benzylamine were applied. Interestingly, when the 1.0 equivalent of benzylamine was used to investigate the potential of sequential ring-opening, only the double ring-opened product 1 was obtained in 32% yield with high stereoselectivity maintained (entry 5, >95:5 dr, 98:2 er).

Subsequent work focused on probing the generality of this process to make a range of substituted malamide derivatives (Figure 2). The capability of alternative primary and secondary amines towards the double ring-opening process was investigated, with good product yields and excellent levels of enantio- and diastereoselectivity (>95:5 dr, >98:2 er) observed with benzylamine, (2-pyridylmethyl)amine and morpholine (to give **1**, **2** and **3**, respectively). The relative configuration within **2** was assigned unambiguously by single crystal X-ray analysis. The absolute configuration was assigned based on the established reactivity of **HyperBTM** in C(1)-ammonium enolate transformations, with all other products assigned by analogy. Inspired by the DuPont Merck MMP inhibitor structure (see Figure 1A), methyl glycinate was employed as a nucleophile, giving the tripeptide mimic malamide **4** in 28% yield with high stereoselectivity (>95:5 dr, 99:1 er). Subsequently, different alkenyl- and aryl-acetic acid-derived homoanhydrides were employed in this protocol using either morpholine or benzylamine as the nucleophilic ring-opening reagent. Electron-rich 4-methoxyphenylacetic anhydride gave the desired product **5** in 63% yield and excellent stereoselectivity (>95:5 dr, 99:1 er). The use of 4-bromophenylacetic anhydride, 3-thienylacetic anhydride and 5-methylhex-3-enoic anhydride were also successfully employed, giving **6**–**8**, respectively, albeit with reduced product yields. While high stereoselectivity was observed for **7** and **8** (>95:5 dr, 97:3 er), for 4-bromophenyl derivative **6**, reduced diastereoselectivity was detected (80:20 dr, 97:3 er), potentially consistent with previous observations of β-lactone epimerisation [115]. Substitution within the isatin scaffold was altered next. The incorporation of an electron-donating 5-MeO-, as well as halogen substituents (5-Br and 6-Cl), were all tolerated, giving **9**–**11** in 79–84% yield and excellent selectivity (≥94:6 dr, 99:1 er). Variation of the N-substituent showed that an *N*-alloc-substituted product was also tolerated, giving **12** in 77% yield, 92:8 dr and >99:1 er.

## 3. Discussion

The proposed mechanism for the formation of malamide derivatives involves the previously reported enantioselective [2+2] cycloaddition strategy that is initiated through attack of the isothiourea Lewis base organocatalyst **HyperBTM** to phenylacetic anhydride, giving acyl ammonium species **13** (Figure 3). Deprotonation by the carboxylate counterion generates the C(1)-ammonium enolate intermediate **14** [36], which, based on previous computational studies with CF_3_-ketones, undergoes a concerted asynchronous [2+2] cycloaddition with *N*-Boc isatin via pre-transition state intermediate **14**. Stereoselectivity is induced by the stereodirecting Ph group derived from the catalyst blocking the approach of the electrophile from the *Re*-face, with the enolate geometry rigidified by a n_O_→σ*_S-C_ chalcogen bond [116,117,118,119,120,121,122,123]. The β-lactol intermediate **15** then collapses to regenerate the isothiourea catalyst **HyperBTM** and the corresponding spirocyclic β-lactone-oxindole **16**. Upon addition of the nucleophilic amine (illustrated using benzylamine), it is assumed that ring-opening of the β-lactone is favoured to initially give **17**. From **17**, two potential pathways to product **1** can be envisaged. Simplistically, ring-opening of **17** may lead directly to **1**. However, a potential explanation for the observation that only double ring-opened malamide products are obtained when using one equivalent of benzylamine, even though an excess of β-lactone is present, could involve in situ intramolecular ring-opening of the oxindole motif from **17**, leading to a reactive pyrrolidindione intermediate **18**, that in turn is ring-opened to generate the observed malamide product **1**. We are currently unable to discriminate between these potential reaction pathways, but both lead to the observed product.

## 4. Materials and Methods

All reagents and solvents were obtained from commercial suppliers and were used without further purification unless otherwise stated. Anhydrides and *N*-Boc isatin were prepared in accordance with the literature [30,124]. Reactions involving moisture-sensitive reagents were carried out in flame-dried glassware under an inert atmosphere (N_2_ or Ar) using standard vacuum line techniques. Anhydrous and ethanol-free CH_2_Cl_2_ was obtained after passing through an alumina column (MBraun SPS-800). Analytical thin layer chromatography (TLC) was performed on pre-coated aluminium plates (Kieselgel 60 F254 silica), and visualisation was achieved using ultraviolet light (254 nm) and/or staining with aqueous KMnO_4_ solution by heating. Manual column chromatography was performed in glass columns fitted with porosity 3 sintered discs over Kieselgel 60 silica using the solvent system stated.

**General procedure:** The appropriate anhydride (1.5 equiv.), protected isatin (1.0 equiv.) and (2*S*,3*R*)-HyperBTM (5 mol%) were dissolved in anhydrous CH_2_Cl_2_ (0.04 M) and cooled to 0 °C. The reaction was started by the addition of *^i^*Pr_2_NEt (1.25 equiv.) and was stirred at 0 °C for 3 h. Then, the appropriate amine (3.0 equiv.) was added, and the mixture was allowed to warm to room temperature to stir overnight. Once complete, the solvent was removed and reduced pressure and the crude reaction mixture was directly subjected to flash silica column chromatography.**Preparation of *tert*-butyl (2-((2*S*,3*S*)-1,4-bis(benzylamino)-2-hydroxy-1,4-dioxo-3-phenylbutan-2-yl)phenyl)carbamate** (**1**): Following the general procedure, phenylacetic anhydride (95.3 mg, 0.38 mmol, 1.5 equiv.), *tert*-butyl 2,3-dioxoindoline-1-carboxylate (61.8 mg, 0.25 mmol, 1.0 equiv.), (2*S*,3*R*)-HyperBTM (3.9 mg, 0.012 mmol, 5 mol%) and *^i^*Pr_2_NEt (54 μL, 0.31 mmol, 1.3 equiv.) in CH_2_Cl_2_ (0.04 M), followed by benzylamine (82 μL, 0.75 mmol, 3.0 equiv.) gave crude product that was purified by flash silica column chromatography (hexane:EtOAc 90:10 to 40:60) to give **1** as a pale yellow glass (91 mg, 77%). ${\left[\alpha \right]}_{D}^{20}$ −127.0 (*c* 2.3 in CHCl_3_); **Chiral HPLC analysis** (Appendix A), Chiralpak IA (95:5 *n*-hexane: *^i^*PrOH, flow rate 2.0 mL·min^−1^, 211 nm and 40 °C) t_R_ (2*S*,3*S*)-**1** 23.8 min, t_R_ (2*R*,3*R*)-**1** 28.1 min and >99:1 er; **IR** ν_max_ (film) 3397, 3316, 3088, 3065, 3030, 2978, 2928, 2249, 1717, 1651, 1639, 1587, 1522, 1497, 1452, 1441, 1391, 1366, 1308, 1231, 1157, 1121, 1045, 1026, 961, 908, 831 and 754; **^1^H NMR** (400 MHz, CDCl_3_) δ_H_: 1.46 (9H, s, OC(C*H*_3_)_3_), 4.25 (1H, dd, *J* 15.1, 5.4, NC*H*_2_), 4.35 (1H, dd, *J* 15.2, 5.5, NC*H*_2_), 4.47 (1H, dd, *J* 15.1, 6.2, NC*H*_2_), 4.50 (1H, dd, *J* 15.2, 6.1, NC*H*_2_), 4.73 (1H, br s, C(3)*H*), 6.86 (1H, dd, *J* 8.0, 7.2, Ar*H*), 7.05 (2H, d, *J* 7.4, Ar*H*), 7.11–7.39 (16H, m, Ar*H* and O*H*), 7.54 (1H, d, *J* 8.0, Ar*H*), 7.99 (1H, s, N*H*), 8.01 (1H, s, N*H*) and 9.15 (1H, br s, N*H*); **^13^C{^1^H} NMR** (101 MHz, CDCl_3_) δ_C_: 28.5 (OC(*C*H_3_)_3_), 43.5 (N*C*H_2_), 43.6 (N*C*H_2_), 55.8 (*C*(2)), 79.4 (O*C*(CH_3_)_3_), 85.0 (*C*(3)H), 122.0 (Ar*C*H), 126.3 (Ar*C*H), 127.3 (Ar*C*H), 127.4 (Ar*C*H), 127.4 (Ar*C*H), 127.5 (Ar*C*H), 128.0 (Ar*C*H), 128.4 (Ar*C*H), 128.7 (Ar*C*H), 128.9 (Ar*C*H), 129.3 (Ar*C*H), 133.5 (Ar*C*), 137.5 (Ar*C*), 137.6 (Ar*C*), 138.8 (Ar*C*), 152.5 (*C*=O), 174.4 (*C*=O) and 175.2 (*C*=O); **HRMS** (ESI^+^) C_25_H_37_N_3_O_5_Na ${\left[M+Na\right]}^{+}$ found 602.2622, requires 602.2625 (−0.5 ppm).**Preparation of *tert*-butyl (2-((2*S*,3*S*)-2-hydroxy-1,4-dioxo-3-phenyl-1,4-bis((pyri-din-2-ylmethyl)amino)butan-2-yl)phenyl)carbamate** (**2**): Following the general procedure, phenylacetic anhydride (127 mg, 0.50 mmol, 1.5 equiv.), *tert*-butyl 2,3-dioxoindoline-1-carboxylate (81.6 mg, 0.33 mmol, 1.0 equiv.), (2*S*,3*R*)-HyperBTM (5.1 mg, 0.017 mmol, 5 mol%) and *i*-Pr_2_NEt (75 μL, 0.43 mmol, 1.3 equiv.) in CH_2_Cl_2_ (0.04 M), followed by 2-picolylamine (104 μL, 1.00 mmol, 3.0 equiv.), gave crude product that was purified by flash silica column chromatography (CH_2_Cl_2_:10% NH_4_OH in MeOH 99:1 to 93:7) to give 2 as a clear glass (142 mg, 73%). ${\left[\alpha \right]}_{D}^{20}$ −138.5 (*c* 1.4 in CHCl_3_); **Chiral HPLC analysis**, Chiralpak IA (80:20 *n*-hexane: *^i^*PrOH, flow rate 1.0 mL·min^−1^, 270 nm, 30 °C) t_R_ (2*R*,3*R*)-2 15.9 min, t_R_ (2*S*,3*S*)-2 18.5 min and 98:2 er; **IR** ν_max_ (film) 3345, 2972, 1722, 1651, 1589, 1518, 1476, 1437, 1366, 1306, 1233, 1157, 1045, 1024 and 951; **^1^H NMR** (500 MHz, CDCl_3_) δ_H_: 1.50 (9H, s, OC(C*H*_3_)_3_), 4.48 (1H, dd, *J* 16.3, 4.9, NC*H*_2_), 4.52 (1H, dd, *J* 16.3, 5.1, NC*H*_2_), 4.60 (1H, dd, *J* 16.5, 5.3, NC*H*_2_), 4.67 (1H, dd, *J* 16.5, 5.9, NC*H*_2_), 4.76 (1H, s, C(3)*H*), 6.90 (1H, ddd, *J* 8.3, 7.2, 1.4, Ar*H*), 7.01 (2H, m, Ar*H*), 7.11–7.20 (9H, m, Ar*H*), 7.23 (1H, d, *J* 7.8, Ar*H*), 7.35 (1H, t, *J* 4.2, N*H*), 7.53 (1H, dd, *J* 8.1, 1.6, Ar*H*), 7.59 (1H, ddd, *J* 7.9, 7.7, 1.9, Ar*H*), 7.62 (1H, ddd, *J* 7.9, 7.8, 1.9, Ar*H*), 7.92 (1H, br s, N*H*), 8.47 (1H, ddd, *J* 4.9, 1.8, 0.9, Ar*H*), 8.52 (1H, ddd, *J* 4.9, 1.8, 1.0, Ar*H*) and 9.22 (1H, br s, N*H*); **^13^C{^1^H} NMR** (126 MHz, CDCl_3_) δ_C_: 28.5 (C(*C*H_3_)_3_), 44.4 (N*C*H_2_), 44.8 (N*C*H_2_), 79.4 (O*C*(CH_3_)_3_), 121.2 (Ar*C*H), 121.6 (Ar*C*H), 122.2 (Ar*C*H), 122.4 (Ar*C*H), 122.6 (Ar*C*), 126.3 (Ar*C*H), 127.8 (Ar*C*H), 128.1 (Ar*C*H), 128.7 (Ar*C*H), 129.3 (Ar*C*H), 133.5 (Ar*C*), 136.7 (Ar*C*H), 136.8 (Ar*C*H), 138.9 (Ar*C*), 148.9 (Ar*C*H), 149.1 (Ar*C*H), 155.4 (Ar*C*), 156.5 (Ar*C*), 174.8 (*C*=O) and 175.0 (*C*=O); **HRMS** (ESI^+^) C_33_H_35_N_5_O_5_Na ${\left[M+Na\right]}^{+}$ found 604.2522, requires 604.2530 (−1.4 ppm).**Preparation of *tert*-butyl (2-((2*S*,3*S*)-2-hydroxy-1,4-dimorpholino-1,4-dioxo-3-phenylbutan-2-yl)phenyl) carbamate** (**3**): Following the general procedure, phenylacetic anhydride (127 mg, 0.50 mmol, 1.5 equiv.), *tert*-butyl 2,3-dioxoindoline-1-carboxylate (81.6 mg, 0.33 mmol, 1.0 equiv.), (2*S*,3*R*)-HyperBTM (5.1 mg, 0.017 mmol, 5 mol%) and *^i^*Pr_2_NEt (75 μL, 0.43 mmol, 1.3 equiv.) in CH_2_Cl_2_ (0.04 M), followed by morpholine (87 μL, 1.0 mmol, 3.0 equiv.), gave a crude product that was purified by flash silica column chromatography (CH_2_Cl_2_:10% NH_4_OH in MeOH 99:1 to 95:5) to give **3** as a clear glass (106 mg, 59%). ${\left[\alpha \right]}_{D}^{20}$ −176.3 (*c* 0.4 in CHCl_3_); **Chiral HPLC analysis**, Chiralpak IA (98:2 *n*-hexane: *^i^*PrOH, flow rate 1.5 mL·min^−1^, 254 nm, 40 °C) t_R_ (2*S*,3*S*)-**3** 10.9 min, t_R_ (2*R*,3*R*)-**3** 21.3 min and 98:2 er; **IR** ν_max_ (film) 3339, 2855, 1722, 1632, 1614, 1587, 1529, 1439, 1366, 1304, 1242, 1227, 1157, 1111 and 1045; **^1^H NMR** (400 MHz, CDCl_3_) δ_H_: 1.41 (9H, s, C(C*H*_3_)_3_), 2.98–3.06 (2H, m, OC*H*_2_), 3.26–3.33 (2H, m, NC*H*_2_), 3.44 (1H, ddd, *J* 11.4, 5.8, 2.9, OC*H*_2_), 3.49–3.67 (7H, m, NC*H*_2_ and OC*H*_2_), 3.71–3.86 (4H, m, NC*H*_2_ and OC*H*_2_), 4.41 (1H, s, C(3)*H*), 6.82 (2H, d, *J* 7.1, Ar*H*), 6.95 (1H, ddd, *J* 8.1, 7.2, 1.3, Ar*H*), 7.08 (1H, dd, *J* 7.9, 1.6, Ar*H*), 7.13–7.22 (4H, m, Ar*H*), 8.03 (1H, d, *J* 8.4, Ar*H*), 8.63 (1H, s, O*H*) and 8.91 (1H, s, N*H*); **^13^C{^1^H} NMR** (101 MHz, CDCl_3_) δ_C_: 28.4 (C(*C*H_3_)_3_), 42.3 (N*C*H_2_), 43.5 (N*C*H_2_), 46.2 (N*C*H_2_), 47.2 (N*C*H_2_), 55.1 (*C*(3)H), 66.0 (O*C*H_2_), 66.5 (O*C*H_2_), 66.6 (O*C*H_2_), 79.2 (O*C*(CH_3_)_3_), 86.6 (*C*(2)), 119.2 (Ar*C*H), 121.4 (Ar*C*H), 123.0 (Ar*C*), 125.7 (Ar*C*H), 128.0 (Ar*C*H), 128.3 (Ar*C*H), 128.9 (Ar*C*H), 129.6 (Ar*C*H), 132.6 (Ar*C*), 139.3 (Ar*C*), 151.9 (*C*=O), 171.8 (*C*=O) and 174.5 (*C*=O); **HRMS** (ESI^+^) C_29_H_37_N_3_O_5_Na ${\left[M+Na\right]}^{+}$ found 562.2515, requires 562.2524 (−1.6 ppm).**Preparation of dimethyl 2,2’-(((2S,3S)-2-(2-((*tert*-butoxycarbonyl)amino)phenyl)-2-hydroxy-3-phenylsuccinyl)bis(azanediyl))diacetate** (**4**): Following the general procedure, phenylacetic anhydride (127 mg, 0.50 mmol, 1.5 equiv.), *tert*-butyl 2,3-dioxoindoline-1-carboxylate (81.6 mg, 0.33 mmol, 1.0 equiv.), (2*S*,3*R*)-HyperBTM (5.1 mg, 0.017 mmol, 5 mol%) and *^i^*Pr_2_NEt (75 μL, 0.43 mmol, 1.3 equiv.) in CH_2_Cl_2_ (0.04 M), followed by glycine methyl ester hydrochloride (126 mg, 1.00 mmol, 3.0 equiv.) and *^i^*Pr_2_NEt (170 μL, 1.00 mmol, 3.0 equiv.), gave a crude product that was purified by flash silica column chromatography (hexane:EtOAc 100:0 to 0:100) to give 4 as a colourless glass (50 mg, 28%). ${\left[\alpha \right]}_{D}^{20}$ −106.9 (*c* 1.5 in CHCl_3_); **Chiral HPLC analysis**, Chiralpak IA (70:30 *n*-hexane: *^i^*PrOH, flow rate 1.0 mL·min^−1^, 254 nm, 30 °C) t_R_ (2*S*,3*S*)-4 6.7 min, t_R_ (2*R*,3*R*)-4 12.7 min and 99:1 er; **IR** ν_max_ (film) 3335, 2978, 1749, 1728, 1655, 1589, 1526, 1443, 1368, 1308, 1234, 1209, 1163, 1047, 1026 and 756; **^1^H NMR** (500 MHz, CDCl_3_) δ_H_: 1.46 (9H, s, OC(C*H*_3_)_3_), 3.69 (3H, s, OC*H*_3_), 3.70 (3H, s, OC*H*_3_), 3.92–4.08 (4H, m, NC*H*_2_), 4.64 (1H, br s, O*H*), 6.72 (1H, br s, N*H*), 6.84 (1H, ddd, *J* 8.2, 7.2, 1.3, Ar*H*), 6.94–7.00 (2H, m, Ar*H*), 7.09–7.19 (4H, m, Ar*H*), 7.37 (1H, br s, Ar*H*), 7.42 (1H, dd, *J* 8.1, 1.5, Ar*H*), 7.89 (1H, br s, N*H*) and 9.01 (1H, br s, N*H*); **^13^C{^1^H} NMR** (126 MHz, CDCl_3_) δ_C_: 28.6 (C(*C*H_3_)_3_), 41.5 (N*C*H_2_), 41.7 (N*C*H_2_), 52.4 (O*C*H_3_), 52.5 (O*C*H_3_), 79.5 (O*C*(CH_3_)_3_), 120.9 (Ar*C*), 122.2 (Ar*C*H), 126.4 (Ar*C*H), 128.1 (Ar*C*H), 128.2 (Ar*C*H), 128.9 (Ar*C*H), 129.4 (Ar*C*H), 133.0 (Ar*C*H), 138.5 (Ar*C*), 152.6 (Ar*C*), 169.5 (*C*=O), 169.6 (*C*=O), 174.7 (*C*=O) and 175.2 (*C*=O); **HRMS** (ESI^+^) C_27_H_33_N_3_O_9_Na ${\left[M+Na\right]}^{+}$ found 566.2108, requires 566.2109 (−0.2 ppm).**Prepration of *tert*-butyl (2-((2*S*,3*S*)-2-hydroxy-3-(*p*-anisyl)-1,4-dimorpholino-1,4-dioxobutan-2-yl) phenyl)carbamate** (**5**): Following the general procedure, *p*-anisylacetic anhydride (157 mg, 0.50 mmol, 1.5 equiv.), *tert*-butyl 2,3-dioxoindoline-1-carboxylate (81.6 mg, 0.33 mmol, 1.0 equiv.), (2*S*,3*R*)-HyperBTM (5.1 mg, 0.017 mmol, 5 mol%) and *^i^*Pr_2_NEt (75 μL, 0.43 mmol, 1.3 equiv.) in CH_2_Cl_2_ (0.04 M), followed by morpholine (87 μL, 1.0 mmol, 3.0 equiv.), gave a crude product that was purified by flash silica column chromatography (hexane:EtOAc 70:30 to 20:80) to give **5** as clear glass (120 mg, 63%). ${\left[\alpha \right]}_{D}^{20}$ −108.0 (*c* 1.4 in CHCl_3_); **Chiral HPLC analysis**, Chiralpak IA (98:2 *n*-hexane: *^i^*PrOH, flow rate 1.5 mL·min^−1^, 254 nm, 40 °C) t_R_ (2*S*,3*S*)-**5** 12.3 min, t_R_ (2*R*,3*R*)-**5** 28.1 min and 99:1 er; **IR** ν_max_ (film) 3337, 2972, 2856, 1722, 1632, 1611, 1589, 1512, 1439, 1366, 1302, 1242, 1159, 1113, 1034 and 910; **^1^H NMR** (400 MHz, CDCl_3_) δ_H_: 1.41 (9H, s, C(C*H*_3_)_3_), 2.99 (1H, ddd, *J* 10.9, 7.5, 2.8, OC*H*_2_), 3.05 (1H, ddd, *J* 11.1, 7.81, 2.9, OC*H*_2_), 3.24–3.34 (2H, m, NC*H*_2_), 3.40–3.64 (10H, m, NC*H*_2_ and OC*H*_2_), 3.72 (3H, s, OC*H*_3_), 3.74–3.86 (2H, m, NC*H*_2_), 4.35 (1H, s, C(3)*H*), 6.64–6.74 (4H, m, Ar*H*), 6.94 (1H, ddd, *J* 8.1, 7.2, 1.3, Ar*H*), 7.07 (1H, dd, *J* 7.9, 1.5, Ar*H*), 7.14–7.22 (1H, m, Ar*H*), 8.07 (1H, d, *J* 8.5, Ar*H*), 8.52 (1H, s, O*H*) and 8.91 (1H, s, N*H*); **^13^C{^1^H} NMR** (101 MHz, CDCl_3_) δ_C_: 28.3 (C(*C*H_3_)_3_), 42.3 (N*C*H_2_), 43.5 (N*C*H_2_), 46.2 (N*C*H_2_), 47.2 (N*C*H_2_), 54.3 (*C*(3)H), 55.0 (O*C*H_3_), 66.0 (O*C*H_2_), 66.1 (O*C*H_2_), 66.5 (O*C*H_2_), 66.6 (O*C*H_2_), 79.2 (O*C*(CH_3_)_3_), 86.6 (*C*(2)), 113.8 (Ar*C*H), 119.2 (Ar*C*H), 121.4 (Ar*C*H), 123.2 (Ar*C*), 124.3 (Ar*C*), 125.8 (Ar*C*H), 128.9 (Ar*C*H), 130.7 (Ar*C*H), 139.4 (Ar*C*), 152.1 (*C*=O), 159.1 (Ar*C*), 171.9 (*C*=O) and 174.8 (*C*=O); **HRMS** (ESI^+^) C_30_H_39_N_3_O_8_Na ${\left[M+Na\right]}^{+}$ found 592.2624, requires 592.2629 (−0.9 ppm).**Preparation of *tert*-butyl (2-((2*S*,3*S*)-1,4-bis(benzylamino)-3-(4-bromophenyl)-2-hydroxy-1,4-dioxobutan-2-yl)phenyl)carbamate** (**6**): Following the general procedure, 2-(4-bromophenyl)acetic anhydride (206 mg, 0.50 mmol, 1.5 equiv.), *tert*-butyl 2,3-dioxoindoline-1-carboxylate (81.6 mg, 0.33 mmol, 1.0 equiv.), (2*S*,3*R*)-HyperBTM (5.1 mg, 0.017 mmol, 5 mol%) and *^i^*Pr_2_NEt (75 μL, 0.43 mmol, 1.3 equiv.) in CH_2_Cl_2_ (0.04 M) followed by benzylamine (109 μL, 1.00 mmol, 3.0 equiv.) gave a crude product that was purified by flash silica column chromatography (hexane:EtOAc 100:0 to 70:30) to give **6** as an inseparable mixture of *anti* and *syn* diastereoisomers (80:20 dr) (88 mg, 41%) as a pale yellow glass. ${\left[\alpha \right]}_{D}^{20}$ −56.2 (*c* 1.3 in CHCl_3_); **Chiral HPLC analysis**, Chiralpak AD-H (70:30 *n*-hexane: *^i^*PrOH, flow rate 1.0 mL·min^−1^, 211 nm, 30 °C) t_R_ (2*R*,3*R*)-**6** 9.7 min, t_R_ (2*S*,3*S*)-**6** 19.7 min and 97:3 er; t_R_ (2*S*,3*R*)-**6** 11.4 min, t_R_ (2*R*,3*S*)-**6** 15.0 min and 77:23 er; **IR** ν_max_ (film) 3312, 3030, 2976, 2930, 2357, 2320, 1717, 1647, 1587, 1522, 1489, 1443, 1366, 1308, 1234, 1161, 1047, 1026 and 1013; **HRMS** (${ESI}^{+}$) C_35_H_36_O_5_N_3_BrNa ${\left[M+Na\right]}^{+}$ found 680.1712, requires 680.1731 (−2.8 ppm); NMR data for major diastereoisomer: **^1^H NMR** (500 MHz, CDCl_3_) δ_H_: 1.46 (9H, s, C(C*H*_3_)_3_), 4.16–4.53 (4H, m, C*H*_2_N), 4.66 (1H, br s, O*H*), 6.77–6.90 (4H, m, Ar*H*), 7.06–7.29 (14H, m, Ar*H*), 7.99 (2H, br s, 2 × N*H*) and 9.12 (1H, br s, N*H*); **^13^C{^1^H} NMR** (126 MHz, CDCl_3_) δ_C_: 28.6 (C(*C*H_3_)_3_), 43.6 (*C*H_2_N), 43.6 (*C*H_2_N), 55.4 (*C*(3)H), 79.7 (*C*(CH_3_)_3_), 86.6 (*C*(2)), 122.3 (Ar*C*H), 126.2 (Ar*C*H), 127.1 (Ar*C*H), 127.3 (Ar*C*H), 127.3 (Ar*C*H), 127.4 (Ar*C*H), 127.6 (Ar*C*H), 127.7 (Ar*C*H), 128.7 (Ar*C*H), 128.8 (Ar*C*H), 129.3 (Ar*C*), 131.0 (Ar*C*H), 131.3 (Ar*C*H), 131.7(Ar*C*), 132.0 (Ar*C*), 132.7 (Ar*C*), 136.9 (Ar*C*), 137.3 (Ar*C*), 137.5 (Ar*C*), 152.5 (*C*=O), 174.4 (*C*=O) and 174.8 (*C*=O); NMR data for minor diastereoisomer: **^1^H NMR** (500 MHz, CDCl_3_) (*selected*) δ_H_: 1.52 (9H, s, C(C*H*_3_)_3_), 3.81–3.88 (1H, m, NC*H*_2_), 4.00–4.07 (1H, m, NC*H*_2_), 4.97–4.97 (1H, m, NC*H*_2_), 6.57–6.62 (2H, m, Ar*H*), 7.34–7.39 (1H, m, Ar*H*), 7.65–7.70 (1H, m, Ar*H*), 7.74–7.78 (1H, m, Ar*H*) and 10.03 (1H, br s, N*H*); **^13^C{^1^H} NMR** (126 MHz, CDCl_3_) (*selected*) δ_C_: 28.7 (C(*C*H_3_)_3_), 43.1 (*C*H_2_N), 43.7 (*C*H_2_N), 80.4 (*C*(CH_3_)_3_), 86.6 (*C*(2)), 122.7 (Ar*C*H), 125.1 (Ar*C*H), 128.7 (Ar*C*H), 133.1 (Ar*C*), 136.2 (Ar*C*), 155.0 (*C*=O), 171.8 (*C*=O) and 172.4 (*C*=O).**Preparation of *tert*-butyl (2-((2*S*,3*S*)-2-hydroxy-1,4-dimorpholino-1,4-dioxo-3-(thiophen-3-yl)butan-2-yl)phenyl) carbamate** (**7**): Following the general procedure, (thiophen-3-yl)acetic anhydride (133 mg, 0.50 mmol, 1.5 equiv.), *tert*-butyl 2,3-dioxoindoline-1-carboxylate (81.6 mg, 0.33 mmol, 1.0 equiv.), (2*S*,3*R*)-HyperBTM (5.1 mg, 0.017 mmol, 5 mol%) and *^i^*Pr_2_NEt (75 μL, 0.43 mmol, 1.3 equiv.) in CH_2_Cl_2_ (0.04 M), followed by morpholine (87 μL, 1.0 mmol, 3.0 equiv.), gave a crude product that was purified by flash silica column chromatography (hexane:EtOAc 50:50 to 10:90) to give **7** as clear glass (73 mg, 40%). ${\left[\alpha \right]}_{D}^{20}$ −151.8 (*c* 0.9 in CHCl_3_); **Chiral HPLC analysis**, Chiralpak IA (98:2 *n*-hexane: *^i^*PrOH, flow rate 1.5 mL·min^−1^, 211 nm, 40 °C) t_R_ (2*S*,3*S*)-**7** 13.1 min, t_R_ (2*R*,3*R*)-**7** 26.5 min and 98:2 er; **IR** ν_max_ (film) 3335, 2857, 1721, 1632, 1614, 1587, 1530, 1439, 1365, 1233, 1157, 1111, 1045 and 995; **^1^H NMR** (400 MHz, CDCl_3_) δ_H_: 1.45 (9H, s, OC(C*H*_3_)_3_), 3.00 (1H, ddd, *J* 10.8, 7.4, 2.8, OC*H*_2_), 3.12 (1H, ddd, *J* 11.1, 7.8, 2.9, OC*H*_2_), 3.26–3.36 (2H, m, NC*H*_2_), 3.41–3.46 (1H, m, OC*H*_2_), 3.50–3.63 (6H, m, NC*H*_2_ and OC*H*_2_), 3.67–3.87 (5H, m, NC*H*_2_ and OC*H*_2_), 4.55 (1H, s, C(3)*H*), 6.43 (1H, dd, *J* 5.0, 1.3, Ar*H*), 6.82 (1H, dd, *J* 3.1, 1.3, Ar*H*), 6.95 (1H, ddd, *J* 8.1, 7.2, 1.3, Ar*H*), 7.05 (1H, dd, *J* 7.9, 1.5, Ar*H*), 7.12 (1H, dd, *J* 4.9, 3.0, Ar*H*), 7.21 (1H, ddd, *J* 8.5, 7.1, 1.5, Ar*H*), 8.12 (1H, d, *J* 8.4, Ar*H*), 8.66 (1H, s, O*H*) and 9.06 (1H, s, N*H*); **^13^C{^1^H} NMR** (101 MHz, CDCl_3_) δ_C_: 28.4 (C(*C*H_3_)_3_), 42.3 (N*C*H_2_), 43.5 (N*C*H_2_), 46.3 (N*C*H_2_), 47.1 (N*C*H_2_), 50.5 (*C*(3)H), 66.0 (O*C*H_2_), 66.1 (O*C*H_2_), 66.5 (O*C*H_2_), 66.6 (O*C*H_2_), 79.4 (O*C*(CH_3_)_3_), 86.3 (*C*(2)), 119.3 (Ar*C*H), 121.5 (Ar*C*H), 123.3 (Ar*C*), 124.5 (Ar*C*H), 125.4 (Ar*C*H), 125.6 (Ar*C*H), 128.3 (Ar*C*H), 129.0 (Ar*C*H), 132.7 (Ar*C*), 139.4 (Ar*C*), 152.3 (*C*=O), 171.6 (*C*=O) and 174.6 (*C*=O); **HRMS** (ESI^+^) C_27_H_35_N_3_O_7_SNa ${\left[M+Na\right]}^{+}$ found 568.2083, requires 568.2088 (−0.9 ppm).**Preparation of *tert*-butyl (2-((2*S*,3*S*,*E*)-1-(benzylamino)-3-(benzylcarbamoyl)-2-hydroxy-6-methyl-1-oxohept-4-en-2-yl)phenyl)carbamate** (**8**): Following the general procedure, (*E*)-5-methylhex-3-enoic anhydride (119 mg, 0.50 mmol, 1.5 equiv.), *tert*- butyl 2,3-dioxoindoline-1-carboxylate (81.6 mg, 0.33 mmol, 1.0 equiv.), (2*S*,3*R*)-HyperBTM (5.1 mg, 0.017 mmol, 5 mol%) and *^i^*Pr_2_NEt (75 μL, 0.43 mmol, 1.3 equiv.) in CH_2_Cl_2_ (0.04 M) followed by benzylamine (109 μL, 1.00 mmol, 3.0 equiv.) gave crude product that was purified by flash silica column chromatography (hexane:EtOAc 100:0 to 70:30) to give 8 (93 mg, 49%) as a colourless solid. **mp** 80–82 °C; ${\left[\alpha \right]}_{D}^{20}$ −89.7 (*c* 0.7 in CHCl_3_); **Chiral HPLC analysis**, Chiralpak IA (97:3 *n*-hexane: *^i^*PrOH, flow rate 1.0 mL·min^−1^, 211 nm, 40 °C) t_R_ (2*S*,3*S*)-8 30.6 min, t_R_ (2*R*,3*R*)-8 34.4 min and 97:3 er; **IR** ν_max_ (film) 3312, 2961, 2928, 2359, 2344, 1732, 1717, 1636, 1589, 1558, 1522, 1437, 1364, 1306, 1234, 1161, 1045, 1026 and 972; **^1^H NMR** (400 MHz, CDCl_3_) δ_H_: 0.75 (3H, d, *J* 6.7, CHC*H*_3_), 0.80 (3H, d, *J* 6.8, CHC*H*_3_), 1.53 (9H, s, C(C*H*_3_)_3_), 2.06–2.16 (1H, m, C*H*(CH_3_)_2_), 4.08–4.17 (1H, m, C(3)*H*), 4.25–4.38 (2H, m, C*H*^A^H^B^N and CH^A^*H*^B^N), 4.46 (1H, dd, *J* 15.1, 6.3, C*H*^A^H^B^N), 4.56 (1H, dd, *J* 15.1, 6.3, CH^A^*H*^B^N), 5.34 (1H, dd, *J* 15.5, 6.5, C(5)*H*), 5.45 (1H, dd, *J* 15.5, 8.7, C(4)*H*), 6.68 (1H, br s, N*H*), 6.93 (1H, t, *J* 7.4, Ar*H*), 7.11–7.16 (2H, m, Ar*H*), 7.21–7.36 (9H, m, Ar*H*), 7.40 (1H, br s, N*H*), 7.57 (2H, d, *J* 8.1, Ar*H*), 8.07 (1H, br s, O*H*) and 9.76 (1H, br s, N*H*); **^13^C{^1^H} NMR** (126 MHz, CDCl_3_) δ_C_: 21.8 (CH*C*H_3_), 22.1 (CH*C*H_3_), 28.6 (C(*C*H_3_)_3_), 31.2 (*C*H(CH_3_)_2_), 43.4 (*C*H_2_N), 43.5 (*C*H_2_N), 53.6 (*C*(3)H), 79.9 (*C*(CH_3_)_3_), 83.1 (*C*(2)OH), 119.4 (*C*(4)H), 121.7 (Ar*C*), 122.6 (2 × Ar*C*H), 126.4 (Ar*C*H), 127.3 (Ar*C*H), 127.4 (Ar*C*H), 127.5 (Ar*C*H), 127.7 (Ar*C*H), 128.6 (Ar*C*H), 128.7 (Ar*C*H), 128.8 (Ar*C*H), 137.5 (Ar*C*), 137.6 (Ar*C*), 137.9 (Ar*C*), 144.5 (*C*(5)H), 153.2 (*C*=O), 174.6 (*C*=O) and 175.5 (*C*=O); **HRMS** (${ESI}^{+}$) C_34_H_41_O_5_N_3_Na ${\left[M+Na\right]}^{+}$ found 594.2928, requires 594.2938 (−1.7 ppm).**Preparation of *tert*-butyl (2-((2*S*,3*S*)-1,4-bis(benzylamino)-2-hydroxy-1,4-dioxo-3-phenylbutan-2-yl)-4-methoxyphenyl)carbamate** (**9**): Following the general procedure, phenylacetic anhydride (127 mg, 0.50 mmol, 1.5 equiv.), *tert*-butyl 5-methoxy-2,3-dioxoindoline-1-carboxylate (91.5 mg, 0.33 mmol, 1.0 equiv.), (2*S*,3*R*)-HyperBTM (5.1 mg, 0.017 mmol, 5 mol%) and *^i^*Pr_2_NEt (75 μL, 0.43 mmol, 1.3 equiv.) in CH_2_Cl_2_ (0.04 M) followed by benzylamine (109 μL, 1.00 mmol, 3.0 equiv.) gave crude product that was purified by flash silica column chromatography (hexane:EtOAc 95:5 to 50:50) to give **9** (160 mg, 80%) as a colourless solid. **mp** 78–80 °C; ${\left[\alpha \right]}_{D}^{20}$ −116.8 (*c* 1.0 in CHCl_3_); **Chiral HPLC analysis**, Chiralpak IA (90:10 *n*-hexane: *^i^*PrOH, flow rate 1.0 mL·min^−1^, 211 nm, 30 °C) t_R_ (2*S*,3*S*)-**9** 29.0 min, t_R_ (2*R*,3*R*)-**9** 46.5 min, >99:1 er; **IR** ν_max_ (film) 3318, 3063, 3030, 2976, 2932, 1719, 1653, 1522, 1454, 1412, 1366, 1288, 1227, 1163, 1042, 1026 and 810; **^1^H NMR** (400 MHz, CDCl_3_) δ_H_: 1.45 (9H, s, C(C*H*_3_)_3_), 3.63 (3H, s, OC*H*_3_), 4.25 (1H, dd, *J* 15.1, 5.4, C*H*^A^H^B^N), 4.34 (1H, dd, *J* 15.1, 5.5, CH^A^*H*^B^N), 4.40–4.53 (2H, m, C*H*_2_N), 4.76 (1H, br s, O*H*), 6.72 (1H, dd, *J* 9.1, 3.0, Ar*H*), 7.05–7.19 (10H, m, Ar*H*), 7.19–7.29 (7H, m, Ar*H*), 7.29–7.38 (1H, m, N*H*), 7.81 (1H, br s, N*H*) and 8.95 (1H, br s, N*H*); **^13^C{^1^H} NMR** (126 MHz, CDCl_3_) δ_C_: 28.6 (C(*C*H_3_)_3_), 43.5 (*C*H_2_N), 43.5 (*C*H_2_N), 55.4 (O*C*H_3_), 77.4 (*C*H), 79.3 (*C*(CH_3_)_3_), 83.6 (*C*OH), 112.0 (Ar*C*H), 113.7 (Ar*C*H), 122.4 (Ar*C*), 127.3 (Ar*C*H), 127.4 (Ar*C*H), 127.5 (Ar*C*H), 127.5 (Ar*C*H), 128.0 (Ar*C*H), 128.2 (Ar*C*H), 128.7 (Ar*C*H), 128.7 (Ar*C*H), 129.4 (Ar*C*H), 131.4 (Ar*C*), 133.5 (Ar*C*), 137.4 (Ar*C*), 137.6 (Ar*C*), 152.9 (*C*=O), 154.8 (Ar*C*), 174.6 (*C*=O) and 175.1 (*C*=O); **HRMS** (${ESI}^{+}$) C_36_H_39_O_6_N_3_Na ${\left[M+Na\right]}^{+}$ found 632.2721, requires 632.2731 (−1.6 ppm).**Preparation of *tert*-butyl (2-((2*S*,3*S*)-1,4-bis(benzylamino)-2-hydroxy-1,4-dioxo-3-phenylbutan-2-yl)-4-bromophenyl)carbamate** (**10**): Following the general procedure, phenylacetic anhydride (127 mg, 0.50 mmol, 1.5 equiv.), *tert*-butyl 5-bromo-2,3-dioxoindoline-1-carboxylate (108 mg, 0.33 mmol, 1.0 equiv.), (2*S*,3*R*)-HyperBTM (5.1 mg, 0.017 mmol, 5 mol%) and *^i^*Pr_2_NEt (75 μL, 0.43 mmol, 1.3 equiv.) in CH_2_Cl_2_ (0.04 M), followed by benzylamine (109 μL, 1.00 mmol, 3.0 equiv.), gave a crude product that was purified by flash silica column chromatography (hexane:EtOAc 100:0 to 70:30) to give **10** as an inseparable mixture of *anti* and *syn* diastereoisomers (94:6 dr) (182 mg, 84%) and as a colourless solid. **mp** 77–79 °C; ${\left[\alpha \right]}_{D}^{20}$ −88.4 (*c* 0.9 in CHCl_3_); **Chiral HPLC analysis**, Chiralpak IA (90:10 *n*-hexane: *^i^*PrOH, flow rate 1.0 mL·min^−1^, 211 nm, 30 °C) t_R_ (2*S*,3*S*)-**10** 12.2 min, t_R_ (2*R*,3*R*)-**10** 30.7 min and 99:1 er; **IR** ν_max_ (film) 3316, 3030, 2978, 2928, 2357, 2320, 1717, 1647, 1454, 1393, 1366, 1300, 1233, 1161, 1049, 1026 and 824; **^1^H NMR** (500 MHz, CDCl_3_) δ_H_: 1.44 (9H, s, C(C*H*_3_)_3_), 4.21–4.44 (3H, m, C*H*_2_N), 4.51 (1H, dd, *J* 15.1, 6.2, C*H*^A^H^B^N), 4.67 (1H, br s, C(3)*H*), 7.04–7.33 (17H, m, Ar*H*), 7.68 (1H, d, *J* 2.4, Ar*H*), 7.90 (1H, br s, N*H*), 8.07 (1H, br s, N*H*) and 9.03 (1H, br s, N*H*); **^13^C{^1^H} NMR** (126 MHz, CDCl_3_) δ_C_: 28.5 (C(*C*H_3_)_3_), 43.5 (*C*H_2_N), 43.6 (*C*H_2_N), 56.5 (*C*(3)H), 79.7 (*C*(CH_3_)_3_), 84.9 (*C*OH), 114.8 (Ar*C*), 121.3 (Ar*C*H), 127.4 (2 × Ar*C*H), 127.6 (Ar*C*H), 127.6 (Ar*C*H), 128.2 (Ar*C*H), 128.4 (Ar*C*H), 128.7 (Ar*C*H), 128.8 (Ar*C*H), 129.4 (Ar*C*H), 131.8 (Ar*C*), 133.2 (Ar*C*), 137.3 (Ar*C*), 137.5 (Ar*C*), 152.2 (*C*=O), 174.0 (*C*=O) and 175.0 (*C*=O); **HRMS** (${ESI}^{+}$) C_35_H_36_O_5_N_3_BrNa ${\left[M+Na\right]}^{+}$ found 680.1723, requires 680.1731 (−1.2 ppm).**Preparation of *tert*-butyl (2-((2*S*,3*S*)-1,4-bis(benzylamino)-2-hydroxy-1,4-dioxo-3-phenylbutan-2-yl)-5-chlorophenyl)carbamate** (**11**): Following the general procedure, phenylacetic anhydride (127 mg, 0.50 mmol, 1.5 equiv.), *tert*-butyl 6-chloro-2,3-dioxoindoline-1-carboxylate (93.0 mg, 0.33 mmol, 1.0 equiv.), (2*S*,3*R*)-HyperBTM (5.1 mg, 0.017 mmol, 5 mol%) and *^i^*Pr_2_NEt (75 μL, 0.43 mmol, 1.3 equiv.) in CH_2_Cl_2_ (0.04 M), followed by benzylamine (109 μL, 1.00 mmol, 3.0 equiv.), gave a crude product that was purified by flash silica column chromatography (hexane:EtOAc 100:0 to 70:30) to give **11** as an inseparable mixture of *anti* and *syn* diastereoisomers (95:5 dr) (160 mg, 79%) and as a colourless solid. **mp** 83–85 °C; ${\left[\alpha \right]}_{D}^{20}$ −109.0 (*c* 0.7 in CHCl_3_); **Chiral HPLC analysis**, Chiralpak ID (90:10 *n*-hexane: *^i^*PrOH, flow rate 1.0 mL·min^−1^, 254 nm, 30 °C) t_R_ (2*S*,3*S*)-**11** 10.1 min, t_R_ (2*R*,3*R*)-**11** 28.6 min and 99:1 er; **IR** ν_max_ (film) 3319, 3063, 3030, 2978, 2930, 1730, 1717, 1653, 1578, 1522, 1454, 1414, 1366, 1281, 1233, 1161, 1051, 1028, 860; **^1^H NMR** (500 MHz, CDCl_3_) δ_H_: 1.46 (9H, s, C(C*H*_3_)_3_), 4.27 (1H, dd, *J* 15.0, 5.5, C*H*^A^H^B^N), 4.36 (1H, dd, *J* 15.1, 5.7, CH^A^*H*^B^N), 4.41–4.52 (2H, m, C*H*_2_N), 4.61 (1H, br s, C*H*), 6.83 (1H, dd, *J* 8.6, 2.2, Ar*H*), 6.99–7.07 (2H, m, Ar*H*), 7.11–7.18 (4H, m, Ar*H*), 7.18–7.36 (10H, m, Ar*H*), 7.44–7.50 (1H, m, Ar*H*), 8.12 (2H, br s, N*H*) and 9.14 (1H, br s, N*H*); **^13^C{^1^H} NMR** (126 MHz, CDCl_3_) δ_C_: 28.5 (C(*C*H_3_)_3_), 43.5 (*C*H_2_N), 43.6 (*C*H_2_N), 56.4 (*C*H), 79.8 (*C*(CH_3_)_3_), 84.8 (*C*OH), 121.8 (Ar*C*H), 127.3 (Ar*C*H), 127.4 (Ar*C*H), 127.6 (Ar*C*H), 127.6 (Ar*C*H), 128.2 (Ar*C*H), 128.4 (Ar*C*H), 128.7 (2 × Ar*C*H), 129.3 (Ar*C*H), 133.2 (Ar*C*), 134.8 (Ar*C*), 137.3 (Ar*C*), 137.5 (Ar*C*), 140.2 (Ar*C*), 152.1 (*C*=O), 174.0 (*C*=O) and 175.0 (*C*=O); **HRMS** (${ESI}^{+}$) C_35_H_36_O_5_N_3_ClNa ${\left[M+Na\right]}^{+}$ found 636.2232, requires 636.2236 (−0.6 ppm).**Preparation of allyl (2-((2*S*,3*S*)-1,4-bis(benzylamino)-2-hydroxy-1,4-dioxo-3-phenylbutan-2-yl)phenyl)carbamate** (**12**): Following the general procedure, phenylacetic anhydride (127 mg, 0.50 mmol, 1.5 equiv.), allyl 2,3-dioxoindoline-1-carboxylate (76.3 mg, 0.33 mmol, 1.0 equiv.), (2*S*,3*R*)-HyperBTM (5.1 mg, 0.017 mmol, 5 mol%) and *^i^*Pr_2_NEt (75 μL, 0.43 mmol, 1.3 equiv.) in CH_2_Cl_2_ (0.04 M) followed by benzylamine (109 μL, 1.00 mmol, 3.0 equiv.), gave a crude product that was purified by flash silica column chromatography (hexane:EtOAc 90:10 to 70:30) to give **12** as an inseparable mixture of *anti* and *syn* diastereoisomers (92:8 dr) (143 mg, 77%) and as a colourless solid. **mp** 73–75 °C; ${\left[\alpha \right]}_{D}^{20}$ −115.6 (*c* 1.0 in CHCl_3_); **Chiral HPLC analysis**, Chiralpak ID (60:40 *n*-hexane: *^i^*PrOH, flow rate 1.0 mL·min^−1^, 254 nm, 40 °C) t_R_ (2*S*,3*S*)-**12** 6.4 min, t_R_ (2*R*,3*R*)-**12** 12.3 min, and >99:1 er; **IR** ν_max_ (film) 3316, 3063, 3030, 1732, 1717, 1647, 1589, 1526, 1447, 1308, 1219 and 1045; **^1^H NMR** (400 MHz, CDCl_3_) δ_H_: 4.26 (1H, dd, *J* 15.2, 5.5, C*H*^A^H^B^N), 4.33 (1H, dd, *J* 15.2, 5.9, CH^A^*H*^B^N), 4.40–4.58 (4H, m, C*H*_2_O and C*H*_2_N), 4.72 (1H, br s, C*H*), 5.22–5.28 (1H, m, CH=C*H*^A^H^B^), 5.31–5.38 (1H, m, CH=CH^A^*H*^B^), 5.88–6.02 (1H, m, C*H*=CH_2_), 6.87–6.93 (1H, m, Ar*H*), 7.00–7.06 (2H, m, Ar*H*), 7.11–7.31 (14H, m, Ar*H*), 7.33–7.46 (1H, m, N*H*), 7.52–7.59 (1H, m, Ar*H*), 7.89–7.96 (1H, m, Ar*H*), 8.09 (1H, br s, N*H*) and 9.38 (1H, br s, N*H*); **^13^C{^1^H} NMR** (126 MHz, CDCl_3_) δ_C_: 43.4 (*C*H_2_N), 43.5 (*C*H_2_N), 56.2 (*C*H), 65.3 (*C*H_2_O), 85.0 (*C*OH), 117.6 (CH=*C*H_2_), 120.6 (Ar*C*H), 122.6 (Ar*C*H), 126.3 (Ar*C*), 127.3 (Ar*C*H), 127.4 (Ar*C*H), 127.5 (Ar*C*H), 127.5 (Ar*C*H), 127.8 (Ar*C*H), 128.3 (Ar*C*H), 128.7 (2 × Ar*C*H), 129.0 (Ar*C*H), 129.3 (Ar*C*H), 133.0 (*C*H=CH_2_), 133.5 (Ar*C*), 137.4 (Ar*C*), 137.6 (Ar*C*), 152.9 (*C*=O), 174.3 (*C*=O) and 175.1 (*C*=O); **HRMS** (${ESI}^{+}$) C_34_H_33_O_5_N_3_Na ${\left[M+Na\right]}^{+}$ found 586.2306, requires 586.2312 (−1.1 ppm).

## 5. Conclusions

In conclusion, this manuscript demonstrates an effective one-pot process for the synthesis of stereodefined functionalised malamides through employing an enantioselective organocatalytic [2+2]-cycloaddition of C(1)-ammonium enolates with isatin derivatives and consecutive double ring-opening of the corresponding β-lactone-spirooxindole. The malamide products can be isolated in moderate to good yields (28% to 84% yield) and excellent enantio- and diastereo-selectivity (up to >95:5 dr, >99:1 er). This route exemplifies a simple and efficient one-pot synthetic protocol to access enantiomerically pure malamide materials using isothioureas as Lewis base catalysts.

## Figures and Tables

**Figure 1 molecules-29-03635-f001:**
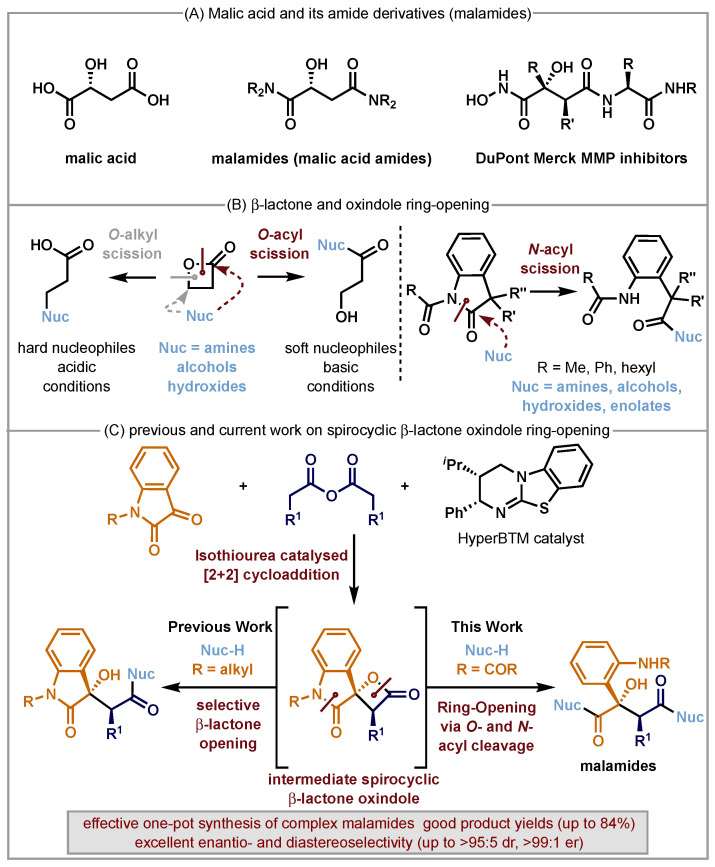
(**A**) Malic acid and malamide derivatives; (**B**) β-lactone and oxindole ring-opening; (**C**) previous and current work on spiro-oxindole-β-lactone ring-opening to generate malamide derivatives.

**Figure 2 molecules-29-03635-f002:**
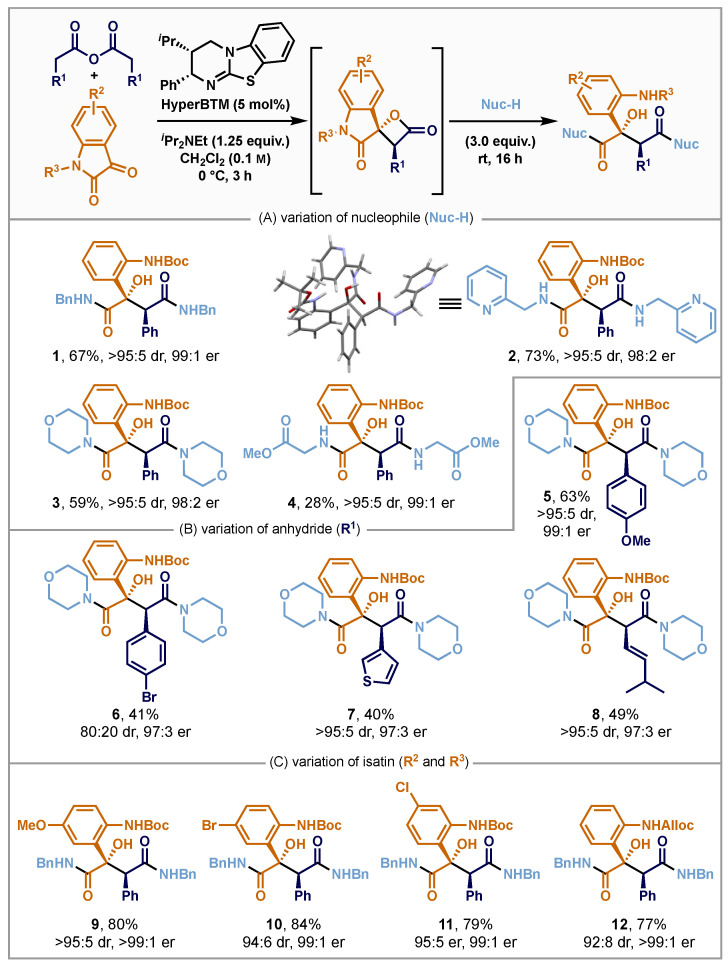
Reaction scope. (**A**) Variation of nucleophile, (**B**) variation of anhydride, (**C**) variation of isatin; typical reaction conditions: The respective isatin (0.50 mmol), homoanhydride (0.33 mmol) and **HyperBTM** (0.02 mmol) were dissolved in CH_2_Cl_2_ (8.5 mL), *^i^*Pr_2_NEt (0.42 mmol) was added, and the reaction was stirred for 3 h at 0 °C. The respective nucleophile (1.00 mmol) was added, and the reaction was stirred overnight at room temperature. Enantiomeric ratios (er) were determined by HPLC analysis on a chiral stationary phase. Diastereomeric ratios (dr) were obtained by ^1^H NMR analysis of the purified reaction product as it could not be determined unambiguously from the crude reaction mixture. Yields are isolated yields.

**Figure 3 molecules-29-03635-f003:**
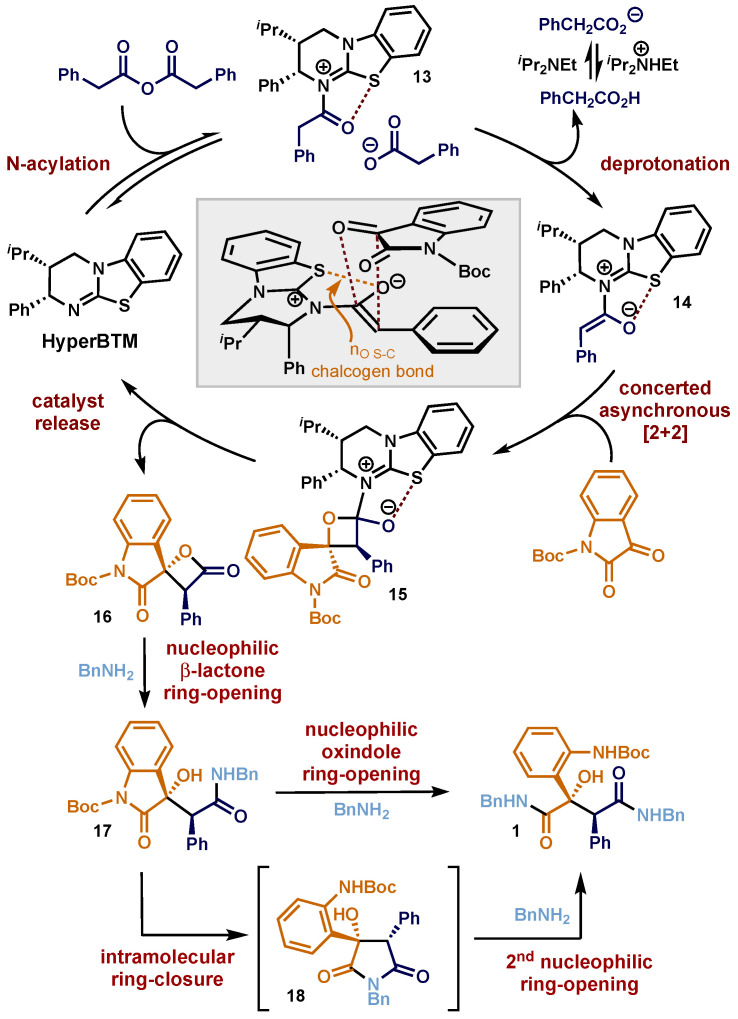
Illustrative proposed reaction mechanism for the generation of stereodefined malamide derivatives.

**Table 1 molecules-29-03635-t001:** Reaction optimisation.

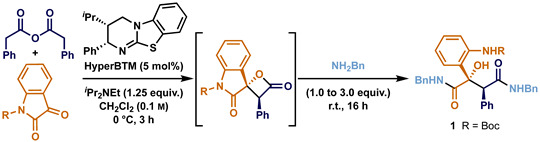				
Entry	R	NH_2_Bn (equiv.)	Product Yield [%] ^b^	dr/er ^c^
1 ^a^	Ph	1.5	0	-
2	C(O)NHPh	3.0	0	-
3	Ts	3.0	Complex mixture	-
4	Boc	3.0	67	>95:5/99:1
5	Boc	1.0	32	>95:5/98:2

^a^ A total of 30% product derived from direct ring-opening of the isatin starting material was observed by analysis of the ^1^H NMR spectra of the crude reaction mixture. ^b^ Yield refers to isolated product yield after chromatographic purification. ^c^ Enantiomeric ratios (er) were determined by HPLC analysis on a chiral stationary phase. Diastereomeric ratios (dr) were obtained by ^1^H NMR analysis of the isolated product mixture.

## Data Availability

All data (experimental procedures and characterisation data) that support the findings of this study are available within the article and its Appendix A. Crystallographic data for compound **2** have been deposited with the Cambridge Crystallographic Data Centre under deposition number 2364405. The research data supporting this publication can be accessed from the St Andrews Research Portal PURE ID 303816895; “One-pot Access to Functionalised Malamides via Organocatalytic Enantioselective Formation and Double Ring-opening of Spirocyclic β-Lactone-Oxindoles” https://doi.org/10.17630/c2137162-53d3-4722-bc51-ae2ee579ce61.

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
