# Peer review of "One-Pot Access to Functionalised Malamides via Organocatalytic Enantioselective Formation of Spirocyclic β-Lactone-Oxindoles and Double Ring-Opening†"

_molecules, 2024, doi:10.3390/molecules29153635_

Round 1

Reviewer 1 Report

Comments and Suggestions for Authors

In this manscript, the authors developed an isothiourea-catalyzed effective one-pot process for the synthesis of functionalised malamides through an enantioselective organocatalytic [2+2]-cycloaddition of (C1)-ammonium enolates with isatin derivatives and followed by consecutive double ring-opening of the corresponding β-lactone-spirooxindoles. The corresponding malamide derivatives can be obtained in moderate to good yields with excellent enantio- and diastereoselectivities. This protocol provide a simple and efficient one-pot synthetic method to access enantiomerically pure malamide derivativess using Lewis base catalysts. In my concern, this paper can be accepted for publication in current form.

Author Response

Review comments: In this manscript, the authors developed an isothiourea-catalyzed effective one-pot process for the synthesis of functionalised malamides through an enantioselective organocatalytic [2+2]-cycloaddition of (C1)-ammonium enolates with isatin derivatives and followed by consecutive double ring-opening of the corresponding β-lactone-spirooxindoles. The corresponding malamide derivatives can be obtained in moderate to good yields with excellent enantio- and diastereoselectivities. This protocol provide a simple and efficient one-pot synthetic method to access enantiomerically pure malamide derivativess using Lewis base catalysts. In my concern, this paper can be accepted for publication in current form.

Author comments: we thank the referee for their positive endorsement of this manuscript - no changes are suggested.

Reviewer 2 Report

Comments and Suggestions for Authors

Author Response

Referee: In the reaction mechanism
proposed, the authors reported two potential ring-opening pathways from
compound 17 to compound 1 (a direct process) or the involvement of
cyclic intermediate 18. Have the authors investigated the double
ring-opening reaction by HRMS analysis?

Author response: We have not investigated the ring-opening reaction by HRMS. While this may give circumstantial evidence of one pathway it will not provide unambiguous determination of the reaction pathway (due to potential cyclisation under the conditions used for mass spectrometry) and so has not been carried out.

Reviewer 3 Report

Comments and Suggestions for Authors

The present research article describes a enantioselective synthesis of functionalized malamides via a one pot sequence that include an organocatalyzed [2+2] cycloaddition of C(1)-ammonium enolate promoted by HyperBTM isothioureas as a key step. The overall strategy relies on 1) the in situ formation of a spirocyclic beta-lactone oxindole intermediate through a [2+2] cycloaddition of C(1)-ammonium enolate (formed by reaction between HyperBTM catalyst and an anhydride) to isatin derivatives followed by 2) a double ring opening of both the oxindole part and the beta-lactone part. Whereas the access to the spirocyclic product starting from for N-alkyl isatins and the nucleophilic ring opening of the b-lactone part has already been described by the same group, the double ring opening of the spirocyclic product (with N-Boc isatins) is original allowing the formation of highly functionalized malamides in high dr and er. Isolated yields are variable ranging from 28% to 84%. The number of examples described is quite limited but sufficient, in my mind, to give a flavor of the field of application of the methodology. The synthetic application of the malamide products could have been pushed a little bit forward especially for the compound 4 that could have been transformed into the known MMP inhibitors described in figure 1A. The proposed mechanism seems to be plausible although the authors couldn’t discriminate the exact pathway for the second ring opening (maybe exhaustive mass spectrometry study could have bring some clues). The spectral data and the quality of the spectra reported in the experimental part and the SI are of high quality.

In conclusion, this is a very interesting publication with high standard results that deserves publication in Molecules after small corrections (see below):

Figure 1 and 3: the beta symbol in beta-lactone appears as a simple b in my PDF version.

In the bottom lines of the table, it is said that the dr were obtained by 1H NMR analyses of the isolated product. Is the authors have checked whether the dr were the same on the crude mixture? Indeed, it is possible to have enrichment of the dr during the purification. If there is no difference between the values before and after purification, please provide the data from the crude mixture. 

Author Response

Reviewer comments: The present research article describes a enantioselective synthesis of functionalized malamides via a one pot sequence that include an organocatalyzed [2+2] cycloaddition of C(1)-ammonium enolate promoted by HyperBTM isothioureas as a key step. The overall strategy relies on 1) the in situ formation of a spirocyclic beta-lactone oxindole intermediate through a [2+2] cycloaddition of C(1)-ammonium enolate (formed by reaction between HyperBTM catalyst and an anhydride) to isatin derivatives followed by 2) a double ring opening of both the oxindole part and the beta-lactone part. Whereas the access to the spirocyclic product starting from for N-alkyl isatins and the nucleophilic ring opening of the b-lactone part has already been described by the same group, the double ring opening of the spirocyclic product (with N-Boc isatins) is original allowing the formation of highly functionalized malamides in high dr and er. Isolated yields are variable ranging from 28% to 84%. The number of examples described is quite limited but sufficient, in my mind, to give a flavor of the field of application of the methodology. The synthetic application of the malamide products could have been pushed a little bit forward especially for the compound 4 that could have been transformed into the known MMP inhibitors described in figure 1A. The proposed mechanism seems to be plausible although the authors couldn’t discriminate the exact pathway for the second ring opening (maybe exhaustive mass spectrometry study could have bring some clues). The spectral data and the quality of the spectra reported in the experimental part and the SI are of high quality.

In conclusion, this is a very interesting publication with high standard results that deserves publication in Molecules after small corrections (see below):

Figure 1 and 3: the beta symbol in beta-lactone appears as a simple b in my PDF version.

Author response: We suggest that this is an error between MAC and PC compatability - the chemdraw figures supplied have the correct formatting and hope this can be corrected at the proof stage.

Referee comments: In the bottom lines of the table, it is said that the dr were obtained by 1H NMR analyses of the isolated product. Is the authors have checked whether the dr were the same on the crude mixture? Indeed, it is possible to have enrichment of the dr during the purification. If there is no difference between the values before and after purification, please provide the data from the crude mixture. 

Author response: We acknowledge the referee's comment that the d.r. can change during chromatographic purification and that normally the dr of the crude material is given in order to facilitate an understadning of the reaction process (this is our usual practice).

However, in the case of these malamide spectra, the crude NMR spectra is too complex to allow for an accurate integration. The spectra of the crude mixtures are not very easy to interpret and there are usually other minor (unknown) impurities obscuring the signals of the product, so assigning d.r. from the crude material is too ambiguous.  Consequently we assigned the dr of the purified material in these instances.

t